# BEYOND DYNAMICS: LEARNING TO DISCOVER CONSERVATION PRINCIPLES

## ABSTRACT

The discovery of conservation principles is crucial for understanding the fundamental behavior of both classical and quantum physical systems across numerous domains. This paper introduces an innovative method that merges representation learning and topological analysis to explore the topology of conservation law spaces. Notably, the robustness of our approach to noise makes it suitable for complex experimental setups and its aptitude extends to the analysis of quantum systems, as successfully demonstrated in our paper. We exemplify our method's potential to unearth previously unknown conservation principles and endorse interdisciplinary research through a variety of physical simulations. In conclusion, this work emphasizes the significance of data-driven techniques in deepening our comprehension of the essential principles governing classical and quantum physical systems.

## 1 INTRODUCTION

Conservation laws and principles lie at the core of our understanding of the physical world, providing insight into the fundamental rules governing the behavior of complex systems. The discovery and application of these principles span various disciplines, such as physics, engineering, biology, and computer science. In recent years, data-driven techniques have emerged as powerful tools for uncovering previously unknown principles and fostering innovative interdisciplinary research.

We present a novel method that tackles the challenge of revealing hidden conservation laws through the synergistic combination of representation learning and topological analysis. Our approach transcends traditional limitations by the robustness to the noise in the measurements, which is important on itself, but also it opens the possibility to work with the quantum systems.

In this paper, we elaborate on the details of our technique, outlining its advantages over existing methods, and demonstrating its potential for uncovering conservation laws in a wide range of settings. By applying our approach to various physical simulations, we showcase its efficacy, broad applicability, and opportunities in interdisciplinary research.

This work aims to further affirm the importance and capability of data-driven research within the landscape of conservation law discovery, contributing to the broader body of knowledge in the field, and advancing our understanding of the intricate systems that define our world.

The paper is organized as follows. In the section 2 *Related work*, we review previous approaches to the discovery of conservation laws, setting the stage for our novel method. The section 3 *Method description* details our data-driven algorithm, which combines representation learning and topological analysis. We report the results of applying our approach to physical simulations in the section 4 *Experiments* and analyze the implications, limitations, and future research in the section 5 *Discussion*. Finally, the section 6 *Conclusion* highlights critical insights and emphasizes the potential of our data-driven technique in advancing our understanding of conservation laws in various disciplines.

## 2 RELATED WORK

Several existing approaches employ data-driven methods for discovering conservation laws of dynamical systems (Liu et al., 2022; Kasim & Lim, 2022; Lejarza & Baldea, 2022; Kaiser et al., 2018; Lu

et al., 2022; Ha & Jeong, 2021; Liu et al., 2023; Iten et al., 2020; Cranmer et al., 2020; Wetzel et al., 2020; Muller, 2022; Mototake, 2021; Liu & Tegmark, 2021; Arora et al., 2023). Specifically, a wide variety of machine learning techniques has been applied in these papers, with examples including the use of conventional neural networks in works such as Liu et al. (2022); Mototake (2021); Ha & Jeong (2021), the utilization of Graph Neural Networks (GNNs) as demonstrated in Cranmer et al. (2020), and the application of Siamese networks in Wetzel et al. (2020), to name but a few.

Some of these methods focus on finding analytical formulas for conserved quantities, which is referred to as symbolic regression in the scientific literature. Discovered symbolic formulas can subsequently enhance the robustness of integration (Channell & Scovel, 1990). It's worth noting that not all approaches consider the physical properties of the data; for instance, in the work of Lample & Charton (2019), the authors do not take into account the data's physical characteristics. In contrast, others, such as Udrescu & Tegmark (2020), introduce symbolic regression techniques tailored to the specific physical domain they are dealing with. These researchers present algorithms designed to uncover symbolic formulas for conserved quantities or dynamic behaviors within the physical context.

Some of the mentioned works exhibit limitations when it comes to the systems they can handle. For instance, certain approaches, such as Liu & Tegmark (2021), exclusively focus on the manifold associated with a single trajectory. Consequently, these methods are ill-suited for systems characterized by a high number of dimensions, typically around 100. In contrast, approaches like those introduced by Ha & Jeong (2021) only have the capacity to learn a solitary conserved quantity, which can pose challenges when dealing with systems featuring numerous conserved quantities.

Moreover, some approaches, such as in Arora et al. (2023), are confined to Hamiltonian systems and cannot accommodate more intricate experiments. Conversely, techniques like those described in Cranmer et al. (2020) are tailored specifically for particular types of systems, such as particle systems. Additionally, in most of the literature, there is a notable absence of consideration for quantum systems, a critical omission given their significance in modern physics.

Finally, the issue of interpretability is a concern for some approaches. For example, Liu & Tegmark (2021) cannot offer meaningful insights for identifying conservation laws beyond estimating the number of them. In contrast, Lu et al. (2022) propose a method capable of uncovering high-dimensional phase spaces and providing correlated quantities for conserved variables.

Our proposed method aims to address these concerns and mitigate the mentioned limitations.

# 3 METHOD DESCRIPTION

## 3.1 PHYSICAL SYSTEM'S DYNAMICS AND CONSERVED QUANTITIES

The subject of our algorithm is dynamical systems. The states of the system are the points in a $d-$dimensional phase space $\mathcal{M}$. For each dynamical system, there is a set (maybe empty) of conserved quantities $H_0(x), \ldots, H_{n-1}(x)$, which are constant functions along each trajectory. Given a single trajectory, all conserved quantities restrict it to the isosurface $\mathcal{M}_h \subseteq \mathcal{M}$, where $h$ is a $n$-dimensional vector representing the values of conserved quantities along the trajectory. The set $\mathcal{C}$ formed by the isosurfaces $\mathcal{M}_h$ is called shape space (Lu et al., 2022). It is a $n$-dimensional manifold since it is parameterized by the $n$-dimensional vector $h$. We will learn the number of conserved quantities in the system by learning the manifold $\mathcal{C}$.

Since our algorithm only has the data from sampled trajectories and does not have any information about the conserved quantities, the manifold $\mathcal{C}$ will be learned from the prospective of $\mathcal{M}_h$, represented by the trajectories. To fully discover $\mathcal{C}$, we should sample the trajectories appropriately, that is, all conserved quantities should variate equally in this set of trajectories; otherwise, the algorithm will not discriminate some directions of the shape space.

Moreover, each trajectory itself should *fully represent* the respective $\mathcal{M}_h$, i.e. the following should hold:

a) Different trajectories with the same conserved quantities $h$ should converge to the unique for the isosurface $\mathcal{M}_h$ physical measure $\mu_h$:

$$\mu_h = \lim_{T \to +\infty} \frac{1}{T} \int\limits_0^T \delta_{x(t)} \mathrm{d}t \tag{1}$$

which means the system should be ergodic (Medio & Lines, 2001).

b) Trajectories should be sampled during a sufficiently large time, such that the trajectory approximates $\mu_h$ well enough (the limit converges).

## 3.2 SHAPE SPACE STRUCTURE ENCODING

To give the Riemannian metric structure to the manifold $\mathcal{C}$ we consider Wasserstein distances (Panaretos & Zemel, 2019) between the distributions $\mu_h$ corresponding to all $\mathcal{M}_h$:

$$W_\beta(\mu_{h_1}, \mu_{h_2}) = \left( \inf_\pi \int ||x_1 - x_2||^\beta \mathrm{d}\pi(x_1, x_2) \right)^{1/\beta} \tag{2}$$

where $\pi$ can be any valid transport map between $\mu_{h_1}$ and $\mu_{h_2}$. Here, $\beta$ is a hyperparameter of the algorithm, which we set to be equal to 2 for all our experiments.

We know the distributions $\mu_h$ from the trajectories. Therefore, we should adjust the formula for the Wasserstein distance for the discrete distributions ($\{x_{11}, \ldots, x_{1m_1}\}$ and $\{x_{21}, \ldots x_{2m_2}\}$) representing two trajectories $x_1$ and $x_2$ sampled at $m_1$ and $m_2$ points correspondingly:

$$W_\beta(\{x_{11}, \ldots, x_{1m_1}\}, \{x_{21}, \ldots x_{2m_2}\}) = \left( \min_T \sum_{i,j} T_{ij} ||x_{1i} - x_{2j}||^\beta \right)^{1/\beta} \tag{3}$$

where $T_{ij}$ meets the following constrains:

$$T_{ij} \geq 0 \; \forall i, j \tag{4}$$

$$\sum_i T_{ij} = 1, \; \sum_j T_{ij} = 1 \; \forall i, j \tag{5}$$

Wasserstein distance should be computed for the dimensionless data. There are infinitely many ways to normalize the trajectories to be dimensionless, resulting in different Wasserstein distances. For our algorithm, such normalization has been chosen that for any $i$, the mean of $i$-th coordinate over all trajectories is 0, and the maximal absolute value of $i$-th coordinate over all trajectories is 1.

## 3.3 METRIC SPACE APPROXIMATION AND PROJECTION (MSAP)

To determine the dimensionality of the shape space $\mathcal{C}$ we look for a Riemannian manifold $\mathcal{C}'$ isometric to the shape space $\mathcal{C}$ while restricting the dimensionality of $\mathcal{C}'$. The resulting $\mathcal{C}'$ will be the `approximation and projection` of the shape space into the $\mathbb{R}^d$. To do that, we first fix a number $l$, numbers $d_1, \ldots, d_l, d$, activation functions $g_1, \ldots, g_l$. Then we try to find the `appropriate` $\mathcal{C}'$ in the set

$$S_k = \{f_\theta^{(k)}(X) \text{ with metric } D_p \mid \forall X \text{ — submanifold of } \mathbb{R}^k, \; \forall \theta \in \mathbb{R}^{m(k)}, \; \forall p > 1\}$$

where $f_\theta^{(k)}$ is a fully connected neural network with the input dimensionality $k$, $l$ intermediate layers with $d_1, \ldots, d_l$ neurons and activations $g_1, \ldots, g_l$, and the output dimensionality $d$; $D_p$ — Minkowsky metric; $m(k)$ is the number of parameters of the $f^{(k)}$. This way we force $S_k$ to contain only manifolds with dimensionality $\leq k$.

If the neural network $f_\theta^{(k)}$ is `large enough`, then the dimensionality of $\mathcal{C}$ will equate to the smallest $k$ that allows us to identify a suitable $\mathcal{C}'$. In numerous instances, the topology of $\mathcal{C}'$ may play a pivotal role in the desired dimensionality. For instance, a circular topology may reside within a 2-dimensional manifold, yet its representation can be effectively condensed to a 1D variable.

Aiding the MSAP in finding an optimal approximation for $\mathcal{C}$ with non-trivial topology can be achieved by seeking approximations across varied $S_k$. For instance, considering $X \subset S^1 \subset \mathbb{R}^2$ compels the MSAP to pinpoint a periodic 1-dimensional manifold. This consideration becomes particularly salient when the shape space exhibits non-trivial topological properties, a phenomenon we elucidate using Turing oscillating patterns as an illustrative example.

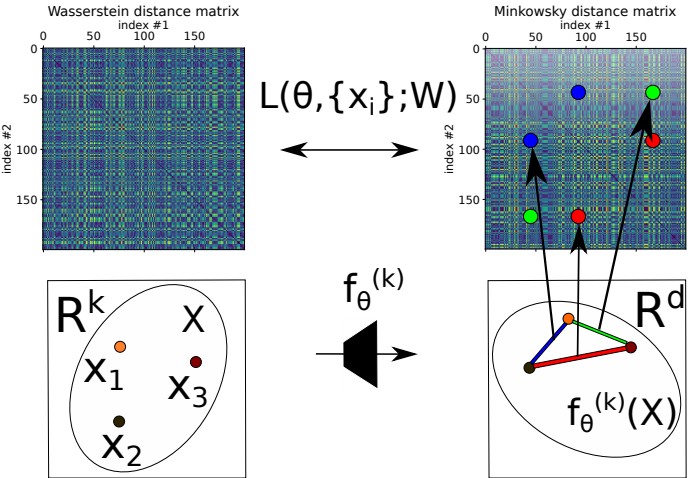

Figure 1: We present in this figure the general idea behind the Metric Space Approximation and Projection. The pairwise Wasserstein distance matrix $W$ is given and has to be approximated. $\{x_j\}$ and $\theta$ are the trainable parameters. To train them we construct the pairwise Minkowsky distance matrix $M$ for $f_\theta^{(k)}(x_j)$ and optimize $L(\theta, \{x_j\}, W) = \text{Stress}(M, W)$.

Note, that $\forall k, \ \forall X$ — submanifold of $\mathbb{R}^k$, $\forall \theta \ \in \ \mathbb{R}^{m(k)}$ one could consider $X' = \{(x_1, \ldots, x_i, 0) | x \in X\} \subset \mathbb{R}^{k+1}$, $\theta'$ — weights for $f_\theta^{(k+1)}$, equal to the weights $\theta$ in the respective coordinates. Then $f_\theta^{(k)}(X) = f_{\theta'}^{(k+1)}(X')$. Therefore, $\forall k : S_k \subseteq S_{k+1}$. The sequence of families $f_\theta^{(k)}$ could be chosen from a more general set than the fully connected neural networks. The important conditions are:

- $f_\theta^{(k)}$ should be continuous, otherwise the resulting set could be a manifold with a dimensionality higher than $k$.
- $\forall k : S_k \subseteq S_k$. The role of this condition is shown in the next paragraph.

However, for simplicity, we will stick with the neural networks described above.

To find the `appropriate` $\mathcal{C}'$ for the given $k$, we minimize the Stress:

$$\{x_i\}, \theta, p = \underset{\substack{x_1, \ldots, x_N \in \mathbb{R}^k, \\ \theta \in \mathbb{R}^{m(k)}, p > 1}}{\arg\min} \text{Stress}(M, W) = \frac{\sum\limits_{i,j}(M_{ij} - W_{ij})^2}{\sum\limits_{i,j} M_{ij}^2}$$

where $W$ – Wasserstein distance matrix, $M$ – Minkowsky distance matrix for the target space: $M_{ij} = D_p\left(f_\theta^{(k)}(x_i), f_\theta^{(k)}(x_j)\right)$.

In the ideal case $L_i > 0$ for $i < n$ and $L_i = 0$ for $i \geq n$. In reality that is improbable, and $L_i > 0 \ \forall i$. Nevertheless, for $i \geq n \ L_i$ will be small. Moreover, since we are trying to approximate the Riemannian manifold $\mathcal{C}$ with the dimensionality $n$, the best approximation also has the dimensionality $n$, and therefore for $i > n \ L_i \geq L_j$. On the other hand, for $i < j \ L_i \geq L_j$ because $\tilde{S}_i \subseteq \tilde{S}_j$. Also, we suppose that for $i < j \leq n \ L_i > L_j$. Therefore, after optimization, we should get the following: $L_1 > L_2 > \cdots > L_n = L_{n+1} = \ldots$.

From the standpoint of the unknown $n$, we got an algorithm to find $n$: get the sequence $L_1, \ldots, L_k$, find the point where it stops decreasing, and the corresponding index will be equal to $n$. Indeed, $k$

should be greater than $n$, but for example, $k$ equal to the number of coordinates in the phase space will be enough.

To reduce the probability of the error, we train each model several times with different initializations and choose the best loss.

### 3.4 MSAP PIPELINE

Combining all parts together, we get the pipeline of the MSAP:

1. The MSAP has several hyperparameters. The most important are the parameters controlling the structure of the neural network: number of layers, number of neurons for each layer, activation functions, target dimensionality.

2. The input data consists of $N$ trajectories with various initial conditions.

3. First step is to normalize the data and compute the pairwise Wasserstein distances between the trajectories.

4. Get the sequence $L_1, \ldots, L_k$ by optimizing the corresponding neural networks and the initial $x_i$.

5. Determine the number of conservation laws by finding the point where the sequence $\{L_i\}$ stops decreasing.

6. As a result of the MSAP we get the determined number of conserved quantities, as well as the points $\{x_i\}$ approximating the shape space.

## 4 EXPERIMENTS

We set up several numerical experiments with dynamical systems with known conserved quantities and tested the proposed method on them. The considered dynamical systems are harmonic oscillator, coupled oscillator, quantum harmonic oscillator and oscillating Turing patterns. We conducted experiments both with noiseless and noisy datasets. In all experiments, dataset consisted of $N = 200$ trajectories. We took the data for the oscillating Turing patterns from Lu et al. (2022). In other experiments we sampled the data theoretically. You can find the further sampling details in the supplementary materials.

### 4.1 COUPLED OSCILLATOR

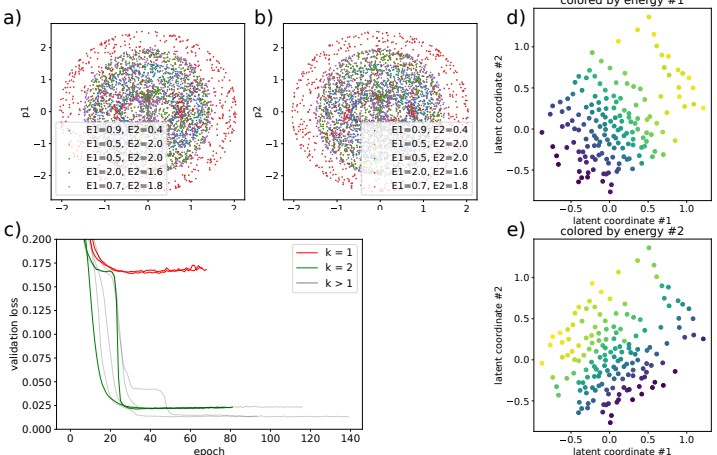

Figure 2: The results of the algorithm applied to the coupled oscillator. (a), (b): A few trajectories of the collected data. (c): Validation learning curves for all $k$ and all random initializations. (d), (e): Graphs showing how the 2d latent coordinates represent the data.

The coupled oscillator consists of 2 massive blocks connected to each other and to the walls by springs (as in the 2a). The coordinates in the coupled oscillator's phase space are the blocks' positions $(x_1, x_2)$ and their momentums $(p_1, p_2)$. Therefore, the phase space of the coupled oscillator is 4-dimensional. The equations of motion of the coupled oscillator are

$$\frac{\mathrm{d}x_1}{\mathrm{d}t} = p_1, \ \frac{\mathrm{d}x_2}{\mathrm{d}t} = p_2, \ \frac{\mathrm{d}p_1}{\mathrm{d}t} = -2x_1 + x_2, \ \frac{\mathrm{d}p_2}{\mathrm{d}t} = -2x_2 + x_1 \tag{6}$$

The coupled oscillator has two conserved quantities, which are energies of two independent modes of oscillating:

$$E_1 = \frac{(x_1 + x_2)^2}{4} + \frac{(p_1 + p_2)^2}{4}, \ E_2 = \frac{3(x_1 - x_2)^2}{4} + \frac{(p_1 - p_2)^2}{4} \tag{7}$$

The figure 2 demonstrates the collected data for the coupled oscillator as well as the results of our algorithm. The validation loss during the training is shown on the graph 2c. We see that the models with $i = 1$ got much worse loss than all other models, which all got the loss approximately $0.04$. That means that the resulting losses stop decreasing at the $i = 2$, so that is the final prediction of our algorithm about the number of the conserved quantities in the system, which is the correct prediction. The graphs on the figures 2d, 2e show how the trained latent representation $X \in \mathbb{R}^2$ represents the trajectories.

## 4.2 HARMONIC OSCILLATOR WITH NOISE

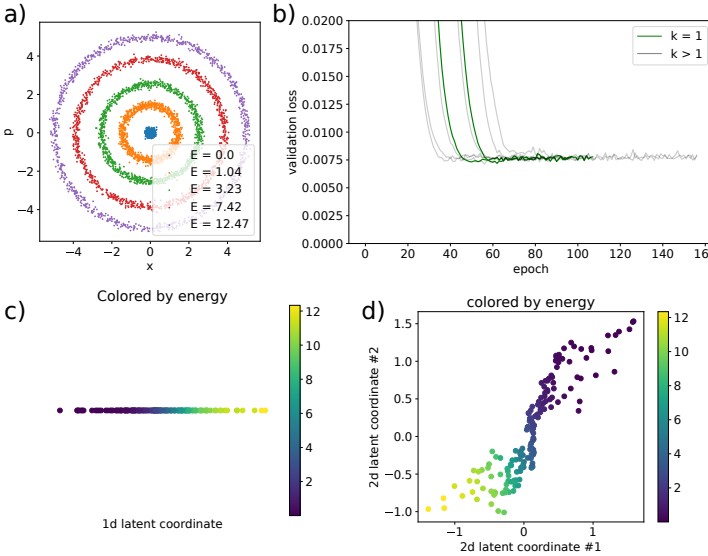

Figure 3: The results of the algorithm applied to the harmonic oscillator. (a): A few trajectories of the collected data. (b): Validation loss curves for all $k$ and all random initializations. (c) Graph showing how the 1d latent coordinates represent the data. (d) Graph showing how the 2d latent coordinates represent the data.

The harmonic oscillator is a simple example of a dynamical system. The harmonic oscillator's phase space consists of two coordinates: coordinate $x$ and momentum $p$. The equations of motion are

$$\frac{\mathrm{d}x}{\mathrm{d}t} = p \ \frac{\mathrm{d}p}{\mathrm{d}t} = -x \tag{8}$$

The harmonic oscillator has one conserved quantity, which is the energy:

$$E = \frac{x^2}{2} + \frac{p^2}{2} \tag{9}$$

The figure 3 demonstrates the collected data for the harmonic oscillator as well as the results of our algorithm. The validation loss during the training is shown on the graph 3b. We see that all models got the loss approximately $0.007 - 0.008$. That means that the resulting losses stop decreasing at the $k = 1$, so that is the final prediction of our algorithm about the number of the conserved quantities in the system, which is the correct prediction. The graph on the figure 3c shows how the trained latent representation $\{x_i\} \in \mathbb{R}^1$ represents the trajectories. Moreover, the figure 3 demonstrates that while training the 2-dimensional representation of the data, model chooses to create a 1-dimensional representation embedded in the 2-dimensional space. This graph shows what happens in dimensions $i > n$.

### 4.3 QUANTUM HARMONIC OSCILLATOR

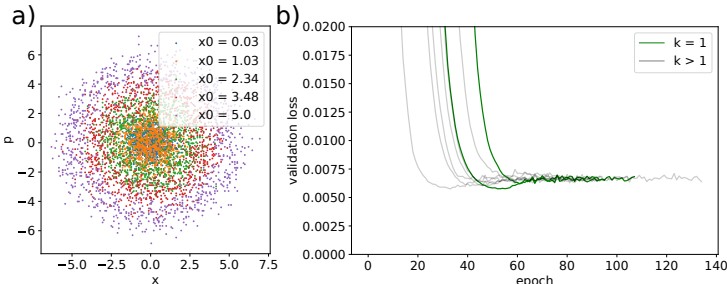

Figure 4: The results of the algorithm applied to the coupled oscillator. (a): A few trajectories of the collected data. (b): Validation learning curves for all $k$ and all random initializations.

To test our approach on a more complicated system, we tested it on the quantum harmonic oscillator. That is a quantum system described by the hamiltonian:

$$\hat{H} = \frac{\hat{p}^2}{2} + \frac{\hat{x}^2}{2} \tag{10}$$

The wavefunction $\psi(x, t)$ evolves according to the Schrodinger equation:

$$\mathrm{i}\frac{\partial \psi}{\partial t} = \hat{H}\psi \tag{11}$$

The total energy $\langle \hat{H} \rangle$ of the quantum harmonic oscillator is its only conserved quantity.

The figure 4 demonstrates the collected data for the quantum harmonic oscillator as well as the results of our algorithm. The validation loss during the training is shown on the graph 4b. We see that for each $i$ best models got the loss approximately $0.04 - 0.05$. That means that the resulting losses stop decreasing at the $i = 1$, so that is the final prediction of our algorithm about the number of the conserved quantities in the system, which is the correct prediction. This example with the quantum harmonic oscillator emphasises the ability of our algorithm to work with the quantum systems, which other algorithms discovering of the number of conserved quantities in the system did not achieve.

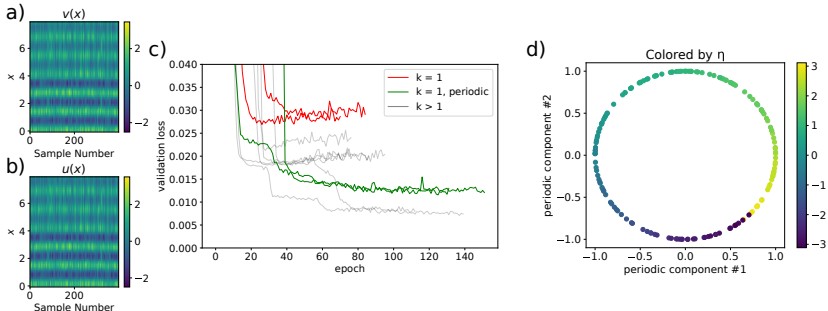

Figure 5: Results of the experiment with oscillating Turing patterns. (a), (b) Collected data. (c) Validation learning curves for all $k$ and all random initializations. (d) Learned periodic embedding.

### 4.4 Oscillating Turing patterns

We consider oscillating Turing patterns as an example of the system with large phase space (dim $\sim 100$). In particular, the Barrio–Varea–Aragón–Maini (BVAM) model is chosen (Barrio, 1999; Aragón et al., 2012):

$$\frac{\partial u}{\partial t} = 0.08 \frac{\partial^2 u}{\partial x^2} + u - v + 1.5uv - uv^2 \tag{12}$$

$$\frac{\partial v}{\partial t} = \frac{\partial^2 v}{\partial x^2} - \frac{3}{4}v + 3u - 1.5uv + uv^2 \tag{13}$$

Aragón et al. (2012) showed that these equations result in oscillating chaotic patterns. There is only one conserved quantity in this system, which is the spatial phase $\eta$ of the patterns that correspond to the position of the pattern in the space. The critical thing about $\eta$ is its periodic topology. To numerically express the phase space, we discretized it on a mesh of size 50, which resulted in the 100-dimensional phase space.

To help our algorithm learn 1-dimensional periodic representation better, we created additional models for $k = 1$ case. To enforce the model to study the periodic embedding, we put the trainable points $\{x_i\}$ on the circle in 2d: $x_1, \ldots, x_N \in S^1 \subset \mathbb{R}^2$, therefore training the periodic 1-dimensional manifolds. Note that this step could be added in the pipeline as well as the similar checks in other dimensions (e.g., for $S^2$).

The figure 5 demonstrates the collected data for the oscillating Turing Patterns as well as the results of our algorithm. The validation loss during the training is shown on the graph 5c. We see that the periodic 1-dimensional model learned approximately as good as the models for larger $k$. That means that the resulting losses stop decreasing at the $k = 1$, so that is the final prediction of our algorithm about the number of the conserved quantities in the system, which is the correct prediction. This example demonstrates that our algorithm can deal with the large phase spaces and also topologically non-trivial shape spaces. Moreover, the figure 5d shows the periodic 1-dimensional representation of the shape space.

## 5 Discussion

### 5.1 Comparison

On the simple example of the coupled oscillator, our approach showed the ability to discover the underlying principles as good as the previous methods, e.g. Iten et al. (2020). However, we achieved the new, previously unmet results for the more complicated systems. More specifically, in the experiment with the oscillating Turing patterns, we showed the ability of our algorithm to uncover topologically non-trivial shape space, while the method proposed by Lu et al. (2022) was only able to represent the same shape space in the higher dimension. Furthermore, our algorithm was able to discover the number of conserved quantities in a quantum system, which has not been done by the previous algorithms.

### 5.2 Limitations

Our approach has a few limitations. One comes from assumptions about the dynamical system and the data collected from it. As mentioned above (see 3.1), the trajectories of the dynamical system should be ergodic. And even though most of the simple physical systems are ergodic, non-ergodic systems exist, which are interesting for discovering but not available for discovery with our approach. Another aspect of data collection is that all conserved quantities should vary. Moreover, variations of different quantities should have approximately the same magnitude. Otherwise, the algorithm will treat the directions with significantly lower variance than those in other directions as noise. In such cases, the algorithm will not detect directions with a small variance, resulting in an underestimated number of conservation laws.

Moreover, systems without conservation laws (for example, pendulum with friction) also cannot be found by our algorithm since the minimal possible prediction about the number of conservation laws for our algorithm is one.

In addition, trajectories should be sampled long enough to represent the respective isosurface well. We need a clearer understanding of determining whether the time in which we sampled the trajectories was enough. However, formulas for the convergence rate of the Wasserstein metric, described by Fournier & Guillin (2014), can help to find the right time to measure or sample the trajectories.

### 5.3 POSSIBLE APPLICATIONS

The proposed method, combining representation learning and topological analysis, has the potential for broad applicability across various domains. Its ability to uncover underlying conservation principles makes it a valuable tool for numerous fields. In robotics and advanced physical system simulation, researchers can utilize the method to optimize energy efficiency, control, and motion planning and improve the accuracy and efficiency of simulations across various physical contexts. Furthermore, this method's applicability extends to control systems, astrophysics, biology, material science, and environmental science. Researchers from these disciplines can leverage this method to understand better the governing principles of the systems they are investigating and drive innovations. Ultimately, this data-driven approach can advance our comprehension of the fundamental tenets of physical systems and beyond.

Notably, the suggested method is computationally inexpensive. Our experiments primarily utilized a single GPU for training. Each of the 1,000 individual MSAP models we prepared for various systems/embeddings took about 10 minutes to train on a single GPU. Including preliminary or failed experiments not reported in the paper, the estimated total computation time was approximately 3000 minutes (50 hours). The developed method's efficiency reduced costs and increased accessibility for researchers implementing similar techniques.

## 6 CONCLUSION

This work introduces an innovative data-driven Metric Space Approximation and Projection approach for discovering unknown conservation laws and principles in complex physical systems. By combining representation learning techniques with topological analysis, our method is able to robustly extract the topology of conservation law spaces, even in the presence of noise.

We have demonstrated the efficacy and versatility of this approach through experiments on various classical and quantum systems. The results showcase the ability of our technique to correctly determine the number of conserved quantities in systems like coupled oscillator and quantum harmonic oscillator. Notably, our method was able to handle experimental noise and uncover the underlying number of conservation laws.

Furthermore, the proposed approach is computationally efficient and does not rely on extensive hyperparameter tuning or expert knowledge. This makes it widely accessible for researchers across disciplines looking to gain insights into the fundamental governing principles of complex systems they are investigating.

Overall, this work emphasizes the potential of data-driven techniques to advance our understanding of conservation laws in classical and quantum systems. The robustness to noise makes our method particularly well-suited for experimental setups. We believe this approach can drive innovations and interdisciplinary collaborations by helping researchers elucidate the core principles that dictate system behavior across numerous domains. Further refinements of the technique could enhance its applicability to even more intricate systems.

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
