# OpenReview forum: "Beyond Dynamics: Learning to Discover Conservation Principles"
_ICLR.cc/2024/Conference — Submitted to ICLR 2024_

### Official Review · Reviewer_dyZf · 2023-10-27

**Soundness:** 2 fair
**Presentation:** 1 poor
**Contribution:** 2 fair
**Rating:** 3
**Confidence:** 3

**Summary:**

The paper talks about a new way to discover conservation laws. By combining representation learning with topological analysis, the method explores the topology of conservation law spaces, showing resilience to noise, and making it apt for complex experimental setups, including quantum systems. The authors tested their method on different scenarios, and it works well, revealing its performance over the prior methods

**Strengths:**

1. The paper proposed MSAP, a new optimal transport based method to discover conservation laws.

2.  The author claimed the benefits of the new method over existing methods.

**Weaknesses:**

1.	The narration of the methodology is not clear. See my questions.

2.	There are a few recent works related to the automatic discovery of conservation laws missing in the literature search:

"Hamiltonian neural networks." Advances in neural information processing systems 32 (2019).

Lagrangian Neural Networks. In ICLR 2020 Workshop on Integration of Deep Neural Models and Differential Equations.

ConCerNet: A Contrastive Learning Based Framework for Automated Conservation Law Discovery and Trustworthy Dynamical System Prediction. International Conference on Machine Learning 2023.

3.	The paper is rushed, with a few incomplete pictures (figure 2, 3, 4 ) and some fonts requires further polishing (figure 1).

4.	Although the method seems to have potential, the presentation quality makes the paper difficult to understand. I suggest some major modifications of the manuscript.

**Questions:**

1.	There are a few places missing notations: equation 1: \delta_{x(t)}, equation 3: T, page 4: all the L_1…L_n are not defined

2.	The definition of S_k in 3.3 is confusing, what does a neural network with metric D_p mean? Does it mean it operates on a space associated with minkowski metric?

3.	Can you explain this sentence in 3.3: “This way we force Sk to contain only manifolds with dimensionality ≤ k”. how this is enforced? And how the Minkowsky metric is involved here?

4.	Can you explain this sentence in 3.3: “then the dimensionality of C will equate to the smallest k that allows us to identify a suitable C ′”. is k a hyperparameter to play with?

5.	Typos: page 4 S_k\inS_k

6. In the latent dynamics model (part v in figure 1), how does the query time t work? Does this model need to solve it iteratively like ODE-like solver to time t or the neural network directly takes t as input and outputs the latent state at time t.

---

### Official Review · Reviewer_vaAm · 2023-10-30

**Soundness:** 3 good
**Presentation:** 2 fair
**Contribution:** 3 good
**Rating:** 5
**Confidence:** 3

**Summary:**

This manuscript presents a data-driven method for discovering hidden conservation laws in physical systems using a combination of representation learning and topological analysis. The approach is robust to noise in measurements and can be applied to both classical and quantum systems. The paper reviews previous approaches to discovering conservation laws and further demonstrates the effectiveness and broad applicability of the proposed method through experiments on physical simulations.

**Strengths:**

- This manuscript introduces a model with the capacity to discover conservation laws in a wide range of physical systems, including quantum systems.
- The proposed method exhibits robustness to measurement noise, rendering it more applicable to real-world scenarios.

**Weaknesses:**

- While the authors emphasize that one of the main advantages of the proposed method is its robustness to noise, they have not provided numerical comparisons with previous approaches to substantiate this claim. I suggest the authors include comparative experiments to support their assertion.
- Additionally, an exploration of how the strength of noise impacts the performance of the proposed method would enhance the comprehensiveness of the study.
- As mentioned by the authors, a limitation of the proposed method is that variations in different quantities should have approximately the same magnitude. Given the importance of this assumption, I suggest the authors introduce it in the Methods section. Additionally, it would be valuable if the authors could provide a specific example to illustrate the failure of the proposed method in cases where this assumption does not hold.

**Questions:**

- See "Weakness" section above.
- In the numerical experiments for quantum systems, the authors describe the data generation process in the supplementary material. The authors state, "To make this setup more realistic, we add some error in the initial condition for each new experiment: both the mean and variance of the Gaussian are normally distributed with means $x_0$ and 1, respectively, and with variances equal to 0.1." I would appreciate clarification on the term "each new experiment." Does it refer to a new initial value $x_0$" for each experiment, or does it means a new measurement conducted on the same $x_0$?
- The authors claim that "we have to repeat the experiment with the same initial conditions as many times as many measurements we want to be made." However, I believe that sampling noise is one of the primary sources of measurement noise in quantum systems. I suggest that the authors delve into a discussion on how the sample complexity impacts the performance of the proposed model.

---

### Official Review · Reviewer_2euw · 2023-10-31

**Soundness:** 1 poor
**Presentation:** 1 poor
**Contribution:** 1 poor
**Rating:** 1
**Confidence:** 2

**Summary:**

n/a

**Strengths:**

n/a

**Weaknesses:**

I do not think this paper is in a suitable form for submission to ICLR. To my mind, a submission to ICLR needs to be accessible to the majority of the audience, and I did not find this paper to be in that form. While the authors' introduction is clear, I found that the subsequent sections of the paper lacked the contextual information to make them comprehensible (I would described myself as not an expert in the area of this paper, but have published in closely related areas). As an example, equation (1) introduces a quantity that seems to be very important to the rest of the paper but does not described what it is or why it is important, and the definition is unclear as it is expressed in terms of \delta_{x(t)} which is not defined anywhere.

**Questions:**

n/a

---

### Official Review · Reviewer_MLvu · 2023-10-31

**Soundness:** 3 good
**Presentation:** 1 poor
**Contribution:** 2 fair
**Rating:** 3
**Confidence:** 5

**Summary:**

This paper proposes a data-driven approach to identify the governing  conservation laws for certain systems. It combines representation learning (by constructing the set S_k) and topological analysis (by introducing appropriate norms and losses). Section 3 is the main contribution of the paper, in section 3.1. some assumptions are introduced, e.g., the dynamical system is ergodic and the trajectories are "long enough" to see most of the phase space, in section 3.2. the Wasserstein distance for the discrete distributions (after normalization for uniqueness of W) is introduced to construct the C as the normed topology, in section 3.3 authors introduce Metric Space Approximation & Projection (MSAP) to find C' as a "reduced" representation of C, for which they use neural networks (although other choices are possible but such choice has the potential to utilize the inverse approximation capability of deep networks). In section 3.4 they also introduce their algorithm. Section 4 examine the application of the algorithm to 4 examples.

**Strengths:**

The ideas behind the construction of C' are a bit rough but potentially novel. Examples, though could be more involved, are illustrating the potentials of the method. Use of stress function as the loss is interesting.

**Weaknesses:**

Two major weaknesses: i) the presentation of the paper could be improved significantly, ii) contribution of the paper needs to be better positioned.

Regarding 1: Section 3.3. needs an overhaul. The assumptions should be clearly written with numbers. The implications of assumptions should be briefly discussed. A theoretical contribution needs necessary for this work: how do we guarantee C' is actually converging to C? Do we have any error analysis (for example when k is equal to the dimension of C, can we say S_k progressively converge to M?
The algorithm needs more clarification than that of page 5. If more space is needed and Appendix can be helpful.

Fig 2 axis are not clear.

Regarding 2: A very clear and distinct set of contributions should be added. Sections 1 and 2 outline the big picture of the paper, but they fail to deep dive into what specific problems are being addressed.
The problem of model discovery has been examined in the past by many researchers (as is also evident from literature review of authors). Each work propose a new direction; its either very suitable for a specific subject, or they are setting up a whole new direction. This paper fails to clearly deliver the message of which direction they are pursuing.
In particular, there are various works that use autoencodrs and they use the latent space to identify an interpretable coordinates to identify the model system in a lower order (see https://arxiv.org/abs/1911.02710, https://www.pnas.org/doi/10.1073/pnas.1906995116, https://www.nature.com/articles/s41598-023-36799-6, among few, where they use concepts from Koopman theory to linearly identify the system, SINDy to symbolically represent the governing equations, and Nueral ODE to parameterize the governing equations). Although authors can argue they look for relatively different concepts (the number of conserved quantities) but all above approaches can also be used for the same task. Plus they have been validated for a long time. They may also be more data efficient and faster. So more comparisons with SOTA is required (also authors should position their work in reference to Hamiltoninan/Lagrangian Neural networks).

**Questions:**

What is the main reason behind choosing L as Stress(M,W)? What would be the outcome of other loss functions are chosen? Some discussion on this choice could be helpful.

How would algorithm work for more complicated dynamical systems, e.g. higher dimensions with chaotic nature or stiffness? Finding conserved quantities in such scenarios can highlight the true power of this algorithm (when compared to other methods).

In abstract and section 1, robustness to noise is mentioned but a very careful analysis for this is missing in the paper.

---

### Meta-Review · Area_Chair_p5Ez · 2023-12-05

**Metareview:**

The paper tries to discover conserved quantities of a physical system from its simulation data. The proposed method is based on representation learning and geometry analysis, and is shown effective in both classical and quantum example systems. Nevertheless, the paper is not written sufficiently accessibly, has major obvious flaws of incomplete results, and differentiation and comparison with other methods are missing.

**Justification For Why Not Higher Score:**

The paper has obvious problems of incomplete results and lacks necessary discussion and comparison with closely related work. The presentation is not friendly (not self-contained and occasionally lacking necessary definitions). The authors did not respond to the problems.

**Justification For Why Not Lower Score:**

N/A

---

### Decision · Program_Chairs · 2024-01-16

Reject